# Radiosensitising Cancer Using Phosphatidylinositol-3-Kinase (PI3K), Protein Kinase B (AKT) or Mammalian Target of Rapamycin (mTOR) Inhibitors

**DOI:** 10.3390/cancers12051278

**Published:** 2020-05-18

**Authors:** Kasun Wanigasooriya, Robert Tyler, Joao D. Barros-Silva, Yashashwi Sinha, Tariq Ismail, Andrew D. Beggs

**Affiliations:** 1College of Medical and Dental Sciences, Institute of Cancer and Genomic Science, University of Birmingham, Edgbaston, Birmingham B15 2TT, UK; j.silva@bham.ac.uk (J.D.B.-S.); yashsinha1991@gmail.com (Y.S.); a.beggs@bham.ac.uk (A.D.B.); 2The New Queen Elizabeth Hospital, University Hospitals Birmingham NHS Foundation Trust, Edgbaston, Birmingham B15 2TH, UK; r.tyler.2@bham.ac.uk (R.T.); tariq.ismail@uhb.nhs.uk (T.I.)

**Keywords:** radiosensitiser, chemoradiotherapy, PI3K, AKT, mTOR inhibitors, rectal cancer, prostate cancer, glioblastoma multiforme, head and neck cancer, non-small cell lung cancer

## Abstract

Radiotherapy is routinely used as a neoadjuvant, adjuvant or palliative treatment in various cancers. There is significant variation in clinical response to radiotherapy with or without traditional chemotherapy. Patients with a good response to radiotherapy demonstrate better clinical outcomes universally across different cancers. The PI3K/AKT/mTOR pathway upregulation has been linked to radiotherapy resistance. We reviewed the current literature exploring the role of inhibiting targets along this pathway, in enhancing radiotherapy response. We identified several studies using in vitro cancer cell lines, in vivo tumour xenografts and a few Phase I/II clinical trials. Most of the current evidence in this area comes from glioblastoma multiforme, non-small cell lung cancer, head and neck cancer, colorectal cancer, and prostate cancer. The biological basis for radiosensitivity following pathway inhibition was through inhibited DNA double strand break repair, inhibited cell proliferation, enhanced apoptosis and autophagy as well as tumour microenvironment changes. Dual PI3K/mTOR inhibition consistently demonstrated radiosensitisation of all types of cancer cells. Single pathway component inhibitors and other inhibitor combinations yielded variable outcomes especially within early clinical trials. There is ample evidence from preclinical studies to suggest that direct pharmacological inhibition of the PI3K/AKT/mTOR pathway components can radiosensitise different types of cancer cells. We recommend that future in vitro and in vivo research in this field should focus on dual PI3K/mTOR inhibitors. Early clinical trials are needed to assess the feasibility and efficacy of these dual inhibitors in combination with radiotherapy in brain, lung, head and neck, breast, prostate and rectal cancer patients.

## 1. Introduction

Radiotherapy is widely used in the treatment of cancer. Radiotherapy in isolation or in combination with chemotherapy may be administered to cancer patients with curative intent [1,2]. Approximately 40% of all patients cured of their cancer will have received radiotherapy as part of their treatment [3]. Radiotherapy can also be used as a neoadjuvant, adjuvant or palliative treatment. Patient response to radiotherapy is widely variable [4,5]. Patients with a good response to radiotherapy generally benefit from a survival advantage over their poorly responding counterparts [4,6]. Those with a good response can also be managed conservatively using “watch and wait” strategies, potentially avoiding the need for further invasive treatment (e.g., surgery) [7,8].

The quest to better understand radiotherapy response and resistance has seen decades of research focus on the intracellular mechanisms activated in response to radiotherapy. Ionising radiation administered as radiotherapy leads to DNA double strand break (DSB) within the cellular genome [9]. This DNA damage activates intracellular pathways geared at either repairing damaged DNA or triggering cell death [10,11]. The two main types of DNA repair processes following DNA DSB include non-homologous end joining (NHEJ) repair and homology directed recombinant (HDR) repair [12,13]. DNA repair through HDR uses template donor DNA from a sister chromatid resulting in a more accurate repair. NHEJ repair joins the damaged ends of the DNA strand together. This can lead to the deletion of DNA segments resulting in changes in the genetic code. Where repair is impossible the cell will undergo apoptosis followed by autophagy. Radiotherapy also promotes significant changes within the tumour microenvironment [14]. The immune system also plays an important role in eliminating damaged or dying cells following radiotherapy through the dispersion of immune system stimulating tumour antigens [15]. Traditionally, 5-fluorauracil (5-FU) based chemotherapy has been used to sensitise tumours to radiotherapy. However, the efficacy of these drugs in patients remains poor [16]. This is arguably due to the clonal selection of chemoradioresistant subpopulations of cells within the treated tumour; harbouring pro-survival and anti-apoptotic mutations [17].

Significant stresses are exerted upon the cell and its microenvironment by radiotherapy. This leads to the activation of pro-survival pathways such as the phosphatidylinositol-3-kinase (PI3K), protein kinase B (AKT) and mammalian target of rapamycin (mTOR) pathway [18]. This complex intracellular pathway contains multiple downstream effectors involved in cell proliferation, growth, survival, cell migration and membrane trafficking (Figure 1 / Appendix A). It is tightly regulated with several activators and negative feedback loops. The PI3K/AKT/mTOR pathway is activated by substrates binding to G-protein coupled receptor (GPCR) or receptor tyrosine kinase (RTK) leading to the activation of membrane bound class-I PI3K proteins [19,20]. Class-I PI3K may also be activated directly by RAS [21]. Activated class-I PI3K phosphorylates transmembrane protein phosphatidylinositol-4,5-bisphosphate (PIP2) to phosphatidylinositol-3,4,5-bisphosphate (PIP3) [22]. This process is inhibited by phosphatase and tensin homolog (PTEN) [23]. PIP3 activates AKT directly by phosphorylation or indirectly via the recruitment of phosphoinositide-dependent protein kinase (PDK1) [24]. mTORC2 is also capable of direct AKT activation [25]. 

Phosphorylated AKT (pAKT) leads to the activation of mTORC1, a principal downstream effector of cell growth, metabolism and protein synthesis [26]. mTORC1 can also be activated by ERK and RSK via the inhibition of tuberous sclerosis proteins (TSC) 1 or 2 [27,28]. Figure 1 demonstrates the complexity of this pathway which has multiple overlapping connections to other intracellular pathways. Activated AKT directly inhibits apoptosis by inhibiting B-cell lymphoma 2 (BCL-2) associated agonist of cell death (BAD), yes-associated protein 1 (YAP), several proteins of the forkhead (FOX) family, BCL-2-associated X protein (BAX), BCL-2 proteins and by activating mouse double minute 2 homolog (MDM2)—an inhibitor of p53 [24]. There is also inhibition of the caspase cell death cascade via the downstream activation of nuclear factor kappa-light-chain-enhancer of activated B cells (NF-KB) or X-linked inhibitor of apoptosis protein (XIAP) by pAKT [29]. There is indirect or direct activation of DNA repair pathway proteins such as DNA-dependent protein kinase (DNA-PK) and proteins of the Fanconi anaemia pathway (FANCD2) by pAKT [30]. AKT activation also leads to cell proliferation, cell cycle progression via the activation of cyclin D1 and inhibition of p27 and inhibitor cyclin dependent kinase [31]. In summary, the cumulative effects of AKT activation promotes cell survival, growth, proliferation, DNA repair and prevents apoptosis.

The PI3K/AKT/mTOR pathway is often dysregulated in malignancy [32]. Mutations within the genes coding for PI3K, PTEN, RAS and epidermal growth factor receptor are frequently detected in cancer [24]. Tumours harbouring these mutations demonstrate radioresistant properties due to the aberrant activation of this pathway [33,34,35,36]. Multiple groups have also demonstrated that the PI3K/AKT/mTOR pathway activation in response to radiotherapy as a principal mechanism of radioresistance [18,37,38,39]. This pathway has also been linked to resistance to chemotherapy [40]. Therefore, the benefits of targeted inhibition of this pathway’s components in the treatment of various cancers has been extensively researched [41]. 

There are numerous pharmacological inhibitors of the PI3K/AKT/mTOR pathway which are widely commercially available (Figure 2). In vitro research has also shown that the inhibition of RAS by EGFR inhibitors leads to enhanced radiation response by indirectly inhibiting the PI3K/AKT/mTOR pathway [42]. Therefore, recent attention has shifted towards direct PI3K/AKT/mTOR pathway component inhibition to enhance radiotherapy response (Table 1, Table 2, Table 3 and Table 4). This literature review identified several studies utilising PI3K, AKT and/or mTOR inhibitors to enhance the radiosensitivity of different types of tumour. Most research investigating the radiosensitising properties of pharmacological inhibitors of the PI3K/AKT/mTOR pathway remains at in vitro or in vivo xenograft stages. A few drugs have undergone early clinical trials (Table 5). The research was broadly categorised based on the type of cancer (adenocarcinoma: Table 1, squamous cell carcinoma (SCC) and non-small cell lung cancer (NSCLC): Table 2, glioblastoma multiforme (GBM): Table 3, and other tumour types: Table 4), and further subcategorised by type of inhibitor (e.g., single or dual).

### 1.1. Sensitising Adenocarcinoma to Radiotherapy

#### 1.1.1. Adenocarcinoma of the Rectum

Neoadjuvant long course chemoradiotherapy (LChRT) followed by surgical resection with total mesorectal excision (TME) is the gold standard treatment for patients with locally advanced rectal cancer (LARC) [43]. Neoadjuvant treatment modalities involving radiotherapy have been shown to significantly reduce the risk of local recurrence and increased rates of clear resection margins (R0 resections) in LARC patients. Complete pathological response (pCR) rates of up to 10–30% and 2–6% have been reported in LARC patients following LChRT or short course radiotherapy (SCRT) respectively [44,45]. Patients with pCR demonstrate a decreased risk of local, distant recurrence and improved survival [45,46]. It is also important to note that there is a significant proportion of rectal cancer patients with a poor response to neoadjuvant therapy [47]. There is an urgent clinical need to improve response to treatment especially in this cohort, given that resistance to chemoradiotherapy poses a significant treatment challenge. 

Research focusing on the use of PI3K/AKT/mTOR pathway inhibitors to improve radiotherapy response in rectal cancer is still in its infancy. In 2015, Chen et al. (1) described the radiosensitising role of dual class-I PI3K and mTORC1/C2 inhibitor (NVP-BEZ235) in vitro using *KRAS* mutant (HCT 116, SW 620) and wild type (HT 29) colorectal (CRC) cell lines [18]. The authors demonstrated activation of the PI3K/AKT/mTOR pathway following radiotherapy in the treated cell lines. There was a gradual increase in the levels of pAKT and phosphorylated mTOR observed up to three hours following exposure to 5 Gy radiotherapy. Dual inhibition with NVP-BEZ235 led to an increase in radiosensitivity of all three CRC lines. The authors observed a reduction in cell proliferation as well as loss of expression of phosphorylated DNA-PKcs and phosphorylated Ribosomal Protein S6 (rpS6) (protein involved in cell growth and proliferation). A significant reduction in HCT 116 CRC mouse xenografts tumour size following NVP-BEZ235 and radiotherapy treatment was also observed (*p* < 0.001). Chen et al. (2) also demonstrated the radiosensitising potential of combined radiotherapy and NVP-BEZ-235 treatment, followed by continued treatment with the drug; in vitro using CRC cell lines HCT 116, HT 29 and SW480, and in vivo using HCT 116 mouse xenografts [48]. This study demonstrated that prolonged treatment after radiotherapy with a dual PI3K/mTOR inhibitor NVP-BEZ235 led to enhanced apoptosis, inhibition of mTOR signalling, prolonged inhibition of cellular viability and disruption of DSB repair pathways. 

Prevo et al. and Djuzenova et al. evaluated the role of PI3K/AKT/DNA-PK inhibitor PI-103 on radiosensitising HCT116, DLD-1 and SW480, SW48 CRC cell lines respectively [23,49]. Djuzenova et al. demonstrated that PI-103 effectively radiosensitised CRC cell lines in the presence of the Heat Shock Protein 90 (HSP-90) inhibitor NVP-AUY922, when both drugs were added three hours before irradiation and continued for 24 h after [49]. A reduction in cell proliferation as well as a rise in level of γ-H2AX was observed. The latter indicating impaired and prolonged DNA DSB repair. The authors also found that any earlier treatment with PI-103 up to 24 h before radiotherapy led to a decline of the radiosensitising effect. The authors concluded that the activation of the MEK/ERK pathway following prolonged inhibition circumvented the PI3K/AKT/mTOR pathway resulting in radioresistance. Prevo et al. demonstrated radiosensitisation independent of EGFR overexpression, *RAS* mutation status and PI3K overexpression in CRC cell lines treated with PI-103 [23]. Reduced pAKT levels, increased G2-M phase delay, delayed DNA DSB repair and cell proliferation contributed to radiosensitisation. The role of isolated mTOR inhibition with everolimus as a radiosensitiser was evaluated by Manegold et al. using CT-26 murine CRC cell lines and an in vivo xenograft mouse model [50]. The authors found that the decreased expression of vascular endothelial growth factor (VEGF) following mTOR inhibition, and the consequent antiangiogenic effects on the tumour microvasculature, was the underlying mechanism of radiosensitivity. This is one of the limited number of studies demonstrating the cancer radiosensitisation following isolated mTOR inhibition.

Two early Phase I/II clinical trials evaluated the role of mTOR inhibitors in conjunction with SCRT in rectal cancer patients (Table 5). Buijsen et al. treated patients with rapamycin one week before and during radiotherapy [51]. The study evaluated the changes in tumour perfusion status and treatment feasibility. This study included thirteen patients in phase I and identified a maximum tolerated dose (MTD) of 6 mg. In phase II, 31 patients were treated with 6 mg of rapamycin and radiotherapy. A higher rate of post-operative complications were observed in phase-I. Consequently, in phase II, surgery was performed after 6 weeks as opposed to 3 days following treatment in phase I. The authors concluded that the combination of SCRT and rapamycin was feasible and safe when using the phase II treatment regimen. There was a significant reduction in tumour metabolic activity on positron emission tomography (PET) scans following treatment (*p* < 0.05). However, no significant increase in pCR was observed in this small sample sized study. Gelsomino et al. also treated 12 patients with 5-FU, radiotherapy and mTOR inhibitor everolimus in another Phase I/II clinical trial [52]. Everolimus was administered 14 days prior to the start of chemoradiotherapy and continued throughout the four-week course of 5FU and radiotherapy. The authors identified a MTD of 10 mg. No significant increase in pCR rates was observed. However, the authors demonstrated that the combination of long course chemoradiotherapy and everolimus was feasible. Further clinical trials are warranted to evaluate the role of mTOR inhibitors and dual pathway inhibitors in improving pCR rates in rectal cancer patients.

#### 1.1.2. Adenocarcinoma of the Prostate

Radiotherapy alone or in combination with chemotherapy and surgery is used in the treatment of localised as well as locally advanced PC respectively [8]. However, PC may recur in approximately 39% of patients who receive radiotherapy for localised disease after a median of 3 years, with an overall mortality rate of 23% at 5 years [53]. In patients with locally advanced PC treated with radiotherapy and docetaxel chemotherapy, 22.7% had a recurrence by 10 years, and the overall 5-year mortality was 41.9% [54]. Recurrence following radiotherapy in local or metastatic PC affects a significant proportion of patients and can be challenging to treat [55]. This is due to the clonal selection and propagation of radioresistant PC cells within these recurrent tumours. There is scope for improving response to radiotherapy in recurrent PC as well as improving response to index radiotherapy treatment in PC patients with targeted therapy. 

Chang et al. evaluated the role of class-I PI3K inhibitor (BKM120), mTORC1 inhibitor (rapamycin), dual class-I PI3K and mTORC1/C2 inhibitors (NVP-BEZ235, PI-103 [also a DNA-PK] inhibitor) in enhancing response to radiotherapy using three radioresistant PC cell lines [56]. The results demonstrated that dual pathway component inhibitors were superior to single component inhibitors in radiosensitising PC cell lines. There was increased apoptosis, autophagy and DNA DSB following the use of dual inhibitors. The authors also observed decreased expression of proteins involved in cell cycle checkpoint inactivation and DNA repair (NHEJ as well as HDR). Potiron et al. and Zhu et al. also demonstrated that NVP-BEZ235 can radiosensitise the PC-3 cell line [57,58]. Potiron et al. also tested the drug on DU145 PC cell line and evaluated the effects of the dual pathway inhibitor on both cell lines under hypoxic and normoxic conditions. In their study, dual inhibition of the pathway successfully radiosensitised both cell lines independent of oxygen concentration or PTEN mutation status [57]. All three studies demonstrated G2-Metaphase (G2-M) arrest within the cell cycle when treated with dual inhibitors.

Diaz et al. studied the effects of isolated AKT inhibitor Palomid 529 on NCI-60 cell line panel in vitro and in vivo using PC-3 xenograft mice [59]. The cell lines were more sensitive to radiotherapy following treatment with the AKT inhibitor. There were reduced levels of pAKT, VEGF and matrix metalloproteinase (MMP)-2, 9. Decreased levels of inhibitor of differentiation/DNA binding protein 1 (Id-1) and BCL-2/BAX protein ratio was also noted. The above proteins are involved in DNA DSB repair, cell survival and proliferation. Further mouse model work demonstrated greater tumour shrinkage (77.4%) after treatment with Palomid 529 and 6 Gy radiotherapy compared to single treatments—(Palomid 529 only—42.9% tumour shrinkage, radiotherapy only 53% tumour shrinkage). There was inhibited cell proliferation and increased apoptosis within the tumour xenograft. Dumont et al. treated mouse xenografts containing PC-3 cells with radioisotope (77Lu) tagged gastrin releasing peptide receptor (GRPr) antagonist, RM2 and mTORC1 inhibitor rapamycin [60]. Combined treatment with 177Lu-RM2 and rapamycin yielded the highest xenograft median survival (82 days), compared to 26 days for untreated mice, and 62 days for radioisotope therapy with GRPr antagonist treatment only group. 

A phase-I clinical trial of 15 patients with locally advanced PC conducted by Azria et al. demonstrated that combined treatment with mTORC1 inhibitor everolimus, hormonal therapy and radiotherapy was feasible [61]. The MTD of everolimus was 7.5 mg/day. However, given the frequency of minor side effects, the authors recommend a treatment dose of 5 mg/day for future studies. In a separate phase-I clinical trial, Narayan et al. used radiotherapy and everolimus to treat patients with a biochemical recurrence of PC following surgery [62]. The authors concluded that this combination was feasible in patients with recurrent PC and that everolimus was well tolerated at doses below 10 mg/day.

#### 1.1.3. Other Adenocarcinoma—Pancreas, breast and from the Gynaecological Tract 

Radiotherapy is often used as an adjuvant or palliative treatment in patients with breast, endometrial, uterine and pancreatic cancer [39,50,63,64,65,66]. Manegold et al. and Park et al. demonstrated radiosensitisation of pancreatic cancer cell lines and pancreatic tumour xenografts following treatment with a mTOR inhibitor (everolimus) and PI3K inhibitor (HS-173) respectively [50,66]. Holler et al. treated breast cancer cell line MDA-MB-231 with mTOR inhibitor Rapamycin. Treatment with the drug alone induced radioresistance in these cells due to the indirect activation of the PI3K/AKT/mTOR pathway components [39]. However, promoting dual pathway inhibition conditions by successful knockdown of AKT (AKT KO) using scramble RNA (sh-RNA) or AKT1-shRNA and rapamycin, radiosensitised MDA-MB-231 cells. There was a significant reduction in cell proliferation and DNA-DSB repair in the AKT KO cells treated with rapamycin and radiotherapy. 

Kuger et al. (1) and Fatehi et al. also tested the dual inhibitor NVP-BEZ235 as a radiosensitiser in breast cancer cell line MDA-MB-231 [63,64]. Fatehi et al. demonstrated that these triple negative breast cancer cells were most radiosensitised when pre-treated with interlukin-6, sirtuin 1 (SIRT1) activator (SRT1720), NVP-BEZ235 [63]. Kuger et al. (1) demonstrated that dual inhibitor NVP-BEZ235 could radiosensitise MDA-MB-231 cells as well as oestrogen receptor positive breast cancer cell line MCF-7, independent of oxygenation status [64]. Research has shown that there are significant variations in the oxygen delivered to the different regions of a solid tumour [67]. Subpopulations of cells within a tumour exposed to different oxygen levels exhibit variable chemoradiotherapy sensitivity [64,68]. The drug radiosensitised hypoxic, normoxic and re-oxygenated cells to the same extent. There was induction of autophagy, delayed DNA DSB repair, inhibited PI3K/mTOR signalling after treatment with the dual inhibitor and radiotherapy, independent of oxygenation status. These findings suggest that pathway inhibition could potentially radiosensitise the heterogeneous population of cells within a tumour.

Miyasaka et al. demonstrated that *TP53* mutation status and PI3K pathway activation contributed to radiotherapy resistance [65]. Nevertheless, the authors successfully radiosensitised endometrial cancer cells containing *TP53* and *PI3K* mutations using the dual PI3K/mTOR inhibitor NVP-BEZ235. The drug in combination with radiotherapy successfully inhibited PI3K/AKT/mTOR signalling which led to reduced cell proliferation and reduced expression of the hypoxia Inducing Factor 1- α (HIF1-α)/VEGF pathway proteins. HIF1-α silencing with small interfering RNA (siRNA) enhanced radiosensitivity further by reducing the sub-G1 cell population. The authors demonstrated the significant relationship between the PI3K/AKT/mTOR pathway and the HIF-α/VEGF pathway in radioresistance, and identified VEGF, HIF-α as potential targets for radiosensitising therapies. 

### 1.2. Sensitising Squamous Cell Carcinoma and Non-Small Cell Lung Cancer to Radiotherapy

#### 1.2.1. Head and Neck Squamous Cell Carcinoma

Approximately half a million patients are diagnosed annually with head and neck (H&N) cancer worldwide [69]. Over 90% of H&N cancer is SCC, originating anywhere along the nasal, para-nasal sinuses, oral cavity or pharyngeal mucosa [70]. Radiotherapy is used in the treatment of H&N SCC in combination with surgery and chemotherapy in the neoadjuvant or adjuvant setting [71]. Prognosis following treatment depends on tumour site and stage [72]. 

We identified several studies evaluating the role of PI3K/AKT/mTOR pathway inhibition radiosensitising H&N SCC (Table 2). Seven of these studies demonstrated the efficacy of dual PI3K and mTOR inhibitors in radiosensitising H&N SCC cell lines [23,37,73,74,75,76,77]. Five of these seven studies (Yu et al., Leiker et al., Liu et al., Cerniglia et al. and Fokas et al. (2) demonstrated radiosensitising effects of dual inhibition in H&N SCC xenograft mice. NVP-BEZ 235 was the most frequently used dual inhibitor. Several other dual PI3K, mTOR inhibitors (PF-05212384, GSK2126458, NVP-BGT226 and PKI-587) yielded similar results [73,74,75,77]. Dual inhibition consistently demonstrated reduced cell proliferation, reduced pAKT, G2-M phase delay and reduced DNA DSB repair characterised by the presence of increased levels of _γ_-H2AX [23,73,74,77]. 

Yu et al. compared the dual inhibitor NVP-BEZ235 against isolated mTOR inhibitors (everolimus and AZD2014) in vitro using H&N SCC cell lines and in vivo using a xenograft model [37]. In this study, dual inhibition was superior to isolated mTOR inhibition, at radiosensitising H&N cell lines. Furthermore, a statistically significant reduction in xenograft tumour size was observed following treatment with a dual inhibitor and radiotherapy (*p* = 0.000017). Impaired AKT/mTOR signalling and G1 phase arrest contributed to radiosensitivity in vitro and in vivo in the dual inhibitor group. Bozec et al. also explored the role of isolated mTOR inhibition in radiosensitising H&N SCC CAL33 mouse xenografts [78]. Treatment with mTOR inhibitor and radiotherapy failed to demonstrate a significant reduction in xenograft tumour size over the control group. However, when combined with traditional chemotherapy, a VEGFA inhibitor (bevacizumab) and radiotherapy, there was significant anti-proliferative effects and a reduction in xenograft tumour size. Isolated mTOR inhibition is therefore unlikely to radiosensitise H&N SCC cells. 

Two early clinical trials evaluating pathway inhibitors as radiosensitisers are currently in progress. A Phase Ib study (ClinicalTrials.gov identifier, NCT02113878) is evaluating the role of BKM120 (PI3K Inhibitor) in combination with cisplatin and radiotherapy in high risk locally advanced H&N SCC. Another Phase I trial is evaluating the feasibility of radiotherapy with everolimus and cisplatin for patients with head and neck cancer (ClinicalTrials.gov identifier, NCT00858663). The results from these trials are pending.

#### 1.2.2. Non-Small Cell Lung Cancer

NSCLC which accounts for over 85% of lung cancer is the leading cause of cancer related death worldwide and may comprise of SCC, adenocarcinoma or other rarer histological subtypes [84]. Radiotherapy is used in the treatment of NSCLC with curative intent, palliative intent or as adjuvant treatment. Outcomes following radiotherapy for NSCLC remains poor even in stage I disease [85]. Numerous studies have explored the role of PI3K/AKT/mTOR pathway inhibitors in radiosensitising NSCLC (Table 2). 

Kim et al. (1) and Konstantinidou et al. tested dual inhibitor NVP-BEZ235 in vitro using NSCLC cell lines, and in vivo using xenograft mice [36,80]. Kim et al. (1) reports a statistically significant reduction in the in vitro NSCLC cell viability following treatment with the dual inhibitor (*p* = 0.01) [80]. The two studies demonstrated radiosensitising effects through increased autophagy, cell proliferation and cell cycle arrest at G1 phase. There was a significant delay in tumour growth observed in the treated xenografts. Additionally, Konstantinidou et al. tried LY294002 with rapamycin, as a dual pathway inhibition treatment [36]. LY294002 is a potent PI3K inhibitor which also weakly inhibits other PI3K-like kinases such as mTOR, ATM and DNA-PK [86]. Using this combination, the authors observed radiosensitising effects which were comparable to NVP-BEZ235 treatment. Toulany et al. also used PI3K, mTOR and DNA-PKcs inhibitor PI-103, and MEK inhibitor PD98059 on an in vitro cell line and in vivo xenograft experiment [38]. The authors demonstrated radiosensitising effects through inhibited DNA DSB repair through suppressed DNA-PKcs activity [38]. In this study, short term PI-103 only treatment led to radiotherapy sensitivity, but long-term treatment for 24 h led to MEK/ERK dependent activation of the PI3K/AKT/mTOR pathway, resulting in radioresistance. 

Kim et al. (2), Kim et al. (3), Mauceri et al. and Holler et al. used mTOR inhibitors (rapamycin or everolimus) on NSCLC cell lines in vitro [39,81,82,83]. Only Mauceri et al. demonstrated isolated mTOR inhibition could lead to radiosensitisation in NSCLC cell lines and xenografts [82]. Holler et al., Kim et al. (2) and Kim et al. (3) found that isolated mTOR inhibition did not radiosensitise NSCLC [39,81,83]. However, Holler et al. demonstrated that when a mTOR inhibitor is combined with an AKT inhibitor (MK2206), significant radiosensitising effects may be observed in NSCLC cell lines [39]. Deutsch et al. showed feasibility of everolimus with radiotherapy in NSCLC patients in an early Phase I clinical trial [87]. Two further clinical trials are currently in progress evaluating everolimus (ClinicalTrials.gov identifier, NCT00374140) and BKM120 (ClinicalTrials.gov identifier, NCT02128724) as radiosensitisers in NSCLC patients.

### 1.3. Sensitising Glioblastoma Multiforme to Radiotherapy

GBM is the most common malignant central nervous system tumour in adults and is routinely treated with chemoradiotherapy with or without surgery [88]. Survival remains poor with studies reporting a median survival of just 14.6 months following chemoradiotherapy [89]. The majority of research exploring the radiosensitising properties of PI3K/AKT/mTOR pathway inhibitors comes from research using GBM cell lines and from a few early clinical trials in GBM patients (Table 3 and Table 5). Djuzenova et al., Del Alcazar et al., Kuger et al. (2), Wang et al., Mukherjee et al. and Cerniglia et al., all demonstrated that dual PI3K/mTOR inhibitors successfully radiosensitise GBM cell lines in vitro [49,74,90,91,92,93]. Del Alcazar et al. and Mukherjee et al. also conducted in vivo research using xenograft mice [90,93]. The most frequently used dual PI3K/mTOR inhibitor was NVP-BEZ235. The drug consistently demonstrated enhanced radiosensitisation of GBM cell lines and in two in vivo mouse xenograft models (Table 3). Two studies using PI3K/mTOR/DNA-PK inhibitor PI-103 also demonstrated enhanced radiosensitisation of GBM cell lines in vitro [49,94]. Dual inhibitors were found to lead to autophagy, apoptosis, G2/M cell cycle arrest and impaired DNA DSB repair. Li et al., Kao et al. and Nakamura et al. tested LY294002 (PI3K and PI3K-like kinase inhibitor) in GBM cell lines [95,96,97]. These studies showed enhanced radiosensitisation of GBM cell lines following treatment with LY294002.

Choi et al., Nakamura et al. and Eshleman et al. tested mTOR inhibitor rapamycin in vitro [94,97,100]. All three studies showed no significant radiosensitising effect of this drug on GBM cell lines. However, Eshelman et al. demonstrated rapamycin enhances radiotherapy sensitivity of U87 spheroids and U87 xenografts [100]. Shinohara et al. studied the effects of mTOR inhibitor everolimus on glioma cells and endothelial cells using in vitro cell line and in vivo mouse model experiments [99]. This study demonstrated the role of mTOR inhibition on impaired tumour blood supply thus leading to inhibition of tumour growth. 

A phase-I trial by Sarkaraia et al. demonstrated that everolimus can be safely administered to GBM patients in conjunction with a chemoradiotherapy treatment regimen comprising of temozolomide and radiotherapy [101]. The authors recommend an everolimus dose of 70 mg/week (10 mg/day) for future studies. However, a phase-II trial by Ma et al. demonstrated that the addition of 70 mg/week of everolimus to standard therapy led to a moderate increase in toxicity with no appreciable benefit in overall survival (*p* = 0.28) or progression free survival (*p* = 0.15) [102]. Another phase-II trial by Chinnaiyan et al. which included 171 patients with GBM found that the addition of everolimus (10 mg/day) to standard therapy led to a significant reduction in overall survival (*p* = 0.008) [103]. Wen et al. conducted a Phase I clinical trial using dual PI3K/mTOR inhibitor XL 765 [104]. The study demonstrated feasibility of the drug in combination with current standard treatment. There was moderate inhibition of the pathway components noted on skin biopsies from patients. The MTD reported by the authors was 90 mg/day.

### 1.4. Sensitising Other Tumour Cells and Endothelial Cells to Radiotherapy

The radiosensitising properties of PI3K/AKT/mTOR inhibitors have also been demonstrated in soft tissue sarcoma, bladder transitional cell carcinoma, hepatocellular carcinoma, metastatic renal cell carcinoma Burkitt’s lymphoma (Table 4), SCC of the cervix (Table 2) and pancreatic cancer (Table 1). Fokas et al. (2) radiosensitised soft tissue sarcoma cell lines and xenograft mice using NVP-BEZ235 [76]. This dual PI3K/mTOR inhibitor led to significant changes within the tumour microenvironment by prolonging the time taken for the normalisation of tumour vascularity. The authors also compared the efficacy of isolated PI3K inhibition using BKM120 against dual inhibition with NVP-BEZ235. The dual inhibitor was a more potent radiosensitiser and led to impaired remodelling of the tumour vasculature [76]. In 2009, Murphy et al. also demonstrated that mTOR inhibitor rapamycin can effectively radiosensitise sarcoma cells in vitro [105].

Prevo et al. demonstrated that PI-103 can successfully radiosensitise sarcoma cell lines and T24 bladder transitional cell carcinoma (TCC) cell lines through cell cycle G1-M phase arrest and inhibited DNA DSB repair [23]. Fokas et al. (1) demonstrated that dual inhibitors (NVP-BEZ235 and NVP-BGT 226) can effectively radiosensitise T24 bladder TCC cell lines and xenograft mice [77]. Assad et al. also demonstrated that mTOR inhibition can effectively radiosensitise cervical SCC cell lines in vitro, at radiotherapy doses as low as 2 Gy [79]. Other types of cancer that may be radiosensitised by pathway inhibition include renal cell carcinoma, hepatocellular carcinoma and Burkitt’s lymphoma [106,107,108]. Given that radiotherapy is less frequently used in the treatment of the aforementioned cancers, the clinical significance of this research is somewhat limited. 

Several studies have also evaluated the role of pathway inhibition in radiosensitising endothelial cells [50,77,99,105]. Resistance to radiotherapy may be induced by tumour revascularisation and inhibiting cellular hypoxia [14]. The pathway inhibitors which sensitise tumour as well as endothelial cells to radiotherapy, suppress the revascularisation of the tumour resulting in an enhanced response to radiotherapy. 

### 1.5. Overcoming Radiotherapy Resistance by Pathway Blockade

Research has identified several biological processes which contribute to the radiosensitisation of cancer, following PI3K/AKT/mTOR pathway inhibition. Most studies demonstrated a reduction in cell proliferation following effective pathway inhibition and irradiation (Table 1, Table 2, Table 3 and Table 4). The majority also reported impaired DNA DSB repair following treatment with pathway inhibitors, characterised by the persistence of excess λ-H2AX foci [18,23,39,49,64,73,74,75]. Several studies demonstrated reduced expression of DNA-PKcs, the protein catalytic subunit of DNA-PK [18,38,66,90,93]. DNA-PK plays an important role in DNA DSB repair [93]. There was enhanced apoptosis and autophagy [56,80,81,83]. Cell cycle progression was inhibited with numerous studies demonstrating cell cycle arrest at G2-M phase following treatment with pathway inhibitors and radiotherapy [56,57,58,66,73,77,78,91,93]. Several others showed cell cycle arrest at G1 phase [23,36,37,65]. The pathway inhibitors successfully led to cell cycle checkpoint activation by inhibiting the inhibitors of cell cycle checkpoint proteins such as TP53. Pathway inhibition also led to significant changes within the tumour microenvironment. There was impaired tumour revascularisation likely through the inhibition of HIF-α and VEGF [65,78]. There was inhibited cell migration [48,73,75]. Pathway inhibitors demonstrated the ability to radiosensitise cells growing in the most hypoxic regions of a tumour, which harbour extreme pro-survival and anti-apoptotic mutations [57,64].

Single pathway inhibition can be ineffective given that the PI3K/AKT/mTOR pathway has numerous triggers (Figure 1) and direct activators of its downstream components, circumventing the effects of this class of inhibitor. Isolated inhibition of mTOR or PI3K likely leads to activation of AKT (the main effector of the pathway) by phosphorylation via the EGFR/RAS/RAF/MEK/ERK pathway [38,49]. AKT inhibition in isolation is also unlikely to be effective because downstream components of the pathway (e.g., mTOR) may be activated via alternative biological pathways [39]. This indirect activation of the PI3K/AKT/mTOR pathway is a crucial mechanism of radiotherapy resistance [38,39,109]. A minimum two-point inhibition of key pathway components is therefore essential to ensure effective downstream inhibition. This may explain the higher efficacy of dual inhibitors such as NVP-BEZ235. The latter was the most frequently used dual PI3K and mTOR inhibitor identified within this type of research. The drug consistently demonstrated radiosensitising effects on in vitro cell line and in vivo xenograft experiments; in PC, CRC, GBM, H&N SCC, NSCLC, breast and gynaecological cancer. Other potential dual or pan pathway inhibitor combinations (e.g., AKT/mTOR or PI3K/mTOR and DNA-PK inhibition) have also been tried [38,39]. However, the current scientific evidence available is limited for making discernible conclusions regarding the radiosensitising potential of these alternative pathway inhibitor combinations.

### 1.6. Limitations, Challenges and Potential for Clinical Translation 

There is significant variation in the methodology used in this research. There were multiple different cell lines being used even within a given tumour type. The majority of this research in rectal cancer utilised cell lines or xenograft models derived from colon cancer [18,23,48,49,50]. However, radiotherapy is only used in the treatment of rectal cancer and not in colon cancer. This remains a significant limitation for rectal cancer research. The advent of novel in vitro experimental models such as three-dimensional primary cell cultures (organoids) using primary rectal tumour tissue, will undoubtedly contribute to novel insights [110]. Variations in radiotherapy regimes used across the different studies was also a significant limitation. Standardisation of radiotherapy regimes preferably at a clinically relevant dose for that tumour type will ensure greater comparability across future research. The duration of treatment with pathway inhibitors varied considerably across the different studies. Even though most authors emphasised the need for pathway inhibitors to be commenced prior to traditional chemoradiotherapy, continued during and preferably for a period thereafter; there remains a lack of consensus on the exact duration of treatment. There was also significant variations in the outcome measures used within this in vitro and in vivo research. These factors render comparability across different studies extremely difficult. 

The challenges of toxicity, the complexity of pathway inhibitor combinations used, as well as the sheer volume of different inhibitors that are available might be hindering translational research in this field (Table 5, Figure 2). The dilemma facing the clinical researcher reviewing the scientific evidence is which inhibitor they must choose for an early clinical trial. For example, we identified at least four different dual PI3K and mTOR inhibitors of this pathway that have been used across the different tumour types. This is a significant limitation in research in this area. Therefore, scientific research should focus on a select few pathway inhibitors and pursue them to clinical trials to ensure effective clinical translation. 

A limited number of pharmacological inhibitors of this pathway have undergone early clinical trials in combination with radiotherapy (Table 5). Several of these early clinical trials used mTOR inhibitors. Frequent side effects of the most commonly used mTOR inhibitor everolimus included mucositis, cutaneous rash and diarrhoea [61,62]. mTOR inhibitors were found to be safe in combination with chemotherapy and/or radiotherapy in cervical cancer, PC and rectal cancer. However, a Phase-II clinical trial from 2018 in GBM patients found overall survival in the treatment group with everolimus to be significantly poorer compared to the standard treatment arm [103]. Most clinical trials using mTOR inhibitors and radiotherapy evaluating pCR rates as an outcome have not shown a clinically significant benefit [51,52]. However, this result should be interpreted with caution given the smaller sample sizes observed in these early clinical trials. Therefore, more phase-II/III multi-arm randomised control trials are needed to make clinically relevant conclusions to this regard.

Only one clinical trial involved a dual PI3K and mTOR inhibitor (Xl-765). The common adverse effects observed with this drug when combined with radiotherapy included nausea, fatigue, thrombocytopenia and diarrhoea [104]. NVP-BEZ235 was the most commonly used dual inhibitor identified in this review. The drug has already been safely used in cancer patients in early clinical trials, as the sole treatment or in combination with other chemotherapy [111]. However, it has yet to undergo an early clinical trial in combination with radiotherapy. The growing evidence from scientific research on the efficacy of dual PI3K and mTOR inhibitors as a radiosensitiser should be the primary focus of current and future scientific research in this area. Clinical trials involving these dual pathway inhibitors and radiotherapy remain scarce. There remains a need for early phase I/II clinical trials using dual PI3K/inhibitors (e.g., NVP-BEZ235) in conjunction with radiotherapy (with or without chemotherapy) in rectal cancer, PC, breast cancer, GBM, H&N SCC and NSCL.

## 2. Conclusions

The PI3K/AKT/mTOR pathway has been identified as a potential radiotherapy resistance inducing pathway in cancer. There is abundant evidence in the current scientific literature which supports the notion that pharmacological inhibition of the PI3K/AKT/mTOR pathway components can radiosensitise cancer. Changing the tumour microenvironment, disrupting DNA DSB repair, inhibiting cell growth, cell proliferation and promoting cell death are a few of the mechanisms by which radiosensitisation may occur. Many pharmacological inhibitors of this pathway have undergone safety and feasibility studies in humans. However, there is limited clinical research available to make reliable conclusions regarding the potency and safety of these inhibitors in radiosensitising tumours in patients.

Research has shown that isolated mTOR or PI3K inhibition can radiosensitise tumours in vitro and in vivo. However, these drugs failed to demonstrate significant clinical efficacy during preliminary trials. Consequently, the attention of researchers has shifted towards the radiosensitising properties of dual PI3K and mTOR inhibitors. Recent scientific research has consistently demonstrated that dual pathway component inhibition to be far superior at radiosensitising different tumour cell types. Therefore, future laboratory as well as clinical research in this area should focus on the translational potential of dual PI3K and mTOR inhibitors as radiosensitisers in cancer patients.

## Figures and Tables

**Figure 1 cancers-12-01278-f001:**
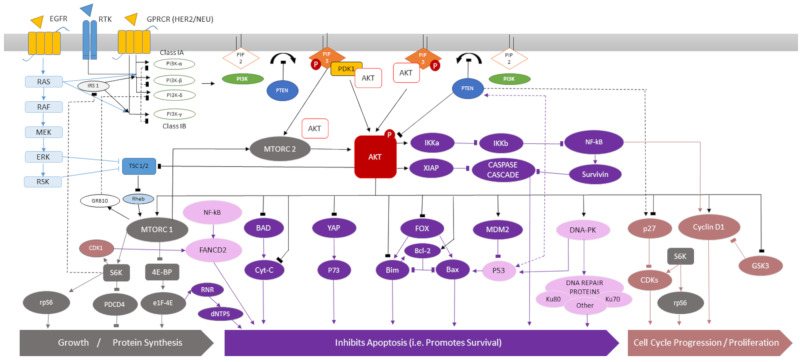
The PI3K/AKT/mTOR pathway activation leads to cell growth, increased protein synthesis, inhibited apoptosis, cell cycle progression and proliferation. For a more detailed description of the pathway components see Appendix A in the supplementary materials section. **Regulatory proteins:** Phosphatidylinositol-3-kinase (PI3K)—class IA (α, β, δ) or class IB (γ), Phosphatidylinositol-4,5-bisphosphate (PIP2), Phosphatidylinositol-3,4,5-bisphosphate (PIP3), 3-phosphoinositide-dependent protein (PDK1), Tuberous sclerosis proteins 1 and 2 (TSC-1/2), RAS homolog enriched in brain (Rheb), Growth factor receptor bound protein 10 (GRB10), Insulin receptor substrate 1 (IRS 1), Phosphatase and tensin homolog (PTEN), Protein kinase B (AKT), Receptor tyrosine kinase (RTK), G-protein coupled receptor (GPRCR), Epidermal growth factor receptor (EGFR). **Proteins involved in cell growth and protein synthesis:** Mammalian target of rapamycin (mTOR), S6 kinase beta-1 (S6K1), Eukaryotic translation initiation factor 4E (eIF4E)-binding protein 1 (4E-BP1), Programmed cell death protein 4 (PDCD4), Ribosomal protein S6 (rpS6). **Proteins involved in promoting cell survival and inhibiting apoptosis:** The IκB kinase alpha (IKKα), IκB kinase beta (IKKβ), nuclear factor kappa-light-chain-enhancer of activated B cells (NF-kB), X-linked inhibitor of apoptosis protein (XIAP), BCL2 associated agonist of cell death (BAD), Cytochrome C (Cyt-C), Yes-associated protein 1 (YAP), p73, Forkhead box proteins (FOX), B-cell lymphoma 2 protein (BCL-2), BCL-2-like protein 11 (Bim), BCL-2-associated X protein (BAX), Mouse double minute 2 homolog (MDM2), DNA-dependent protein kinase (DNA-PK), Ku-80, Ku-70 Ribonucleotide reductase (RNR), Deoxynucleoside triphosphate (dNTP), Survivin, Caspase cascade proteins. **Proteins involved in cell cycle progression and proliferation:** Cyclin dependent kinase 1 (Cyclin D1), Glycogen synthase kinase 3 (GSK3), p27, Cyclin dependent kinases (CDKs).

**Figure 2 cancers-12-01278-f002:**
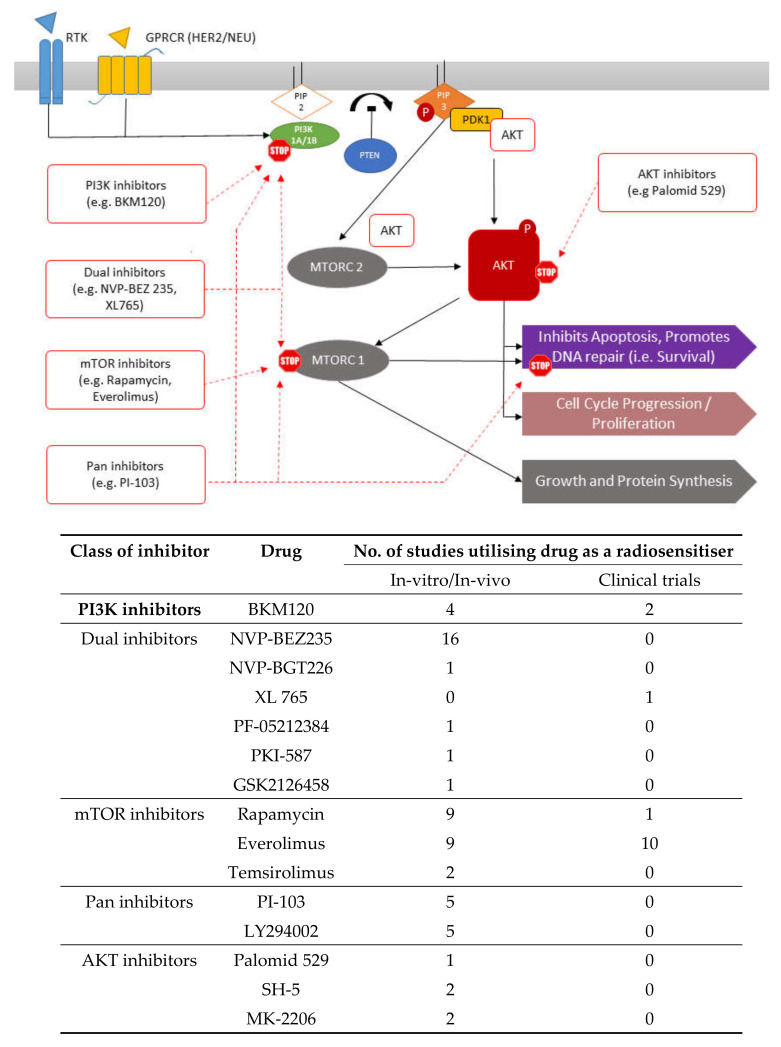
Targets for inhibition by various inhibitors along the PI3K/AKT/mTOR pathway.

**Table 1 cancers-12-01278-t001:** Preclinical studies exploring the role of PI3K, AKT and/or mTOR inhibitors radiosensitising adenocarcinoma.

Author	Year	Type of Cancer	Experimental Model	Drug (s) Tested	Drug Category	Radiotherapy Dose	Summary Outcome
Fatehi et al. [63]	2018	Breast	In vitro (cell line): MDA-MB231	NVP-BEZ235	PI3K and mTOR inhibitor	2 Gy (Gamma radiation)	NVPBEZ235 related radiotherapy sensitivity significantly increased by IL-6 pre-treatment followed by exposure to sirtuin 1 inhibitor SRT1720 (*p* < 0.05)
Holler et al. [39]	2016	Breast	In vitro (cell line): MDA-MB-231	Rapamycin	mTOR inhibitor	Variable range (0–5 Gy)	Rapamycin induced radioresistance in MDA-MB-231 breast cancer cells AKT knockdown by scramble sh-RNA or AKT1-shRNA leads to statistically significant Rrapamycin induced radiosensitisation
Kuger et al.(1) [64]	2014	Breast	In vitro (cell lines): MCF-7, MDA-MB-231	NVP-BEZ235	PI3K and mTOR inhibitor	Variable range (0–8 Gy)	Radiosensitisation observed independent of hypoxia Depleted levels of pAKT, inhibited HIF-1α expression, PI3K/mTOR signalling There was autophagy induction and delayed DNA repair
Miyasaka et al. [65]	2015	Endometrial	In vitro (cell lines): HEC-108, HEC-6, HEC-151, Ishikawa, HEC-59, HEC-50B,HEC-1B, HEC-116	NVP-BEZ235	PI3K and mTOR inhibitor	Variable range (2–6 Gy)	Suppression of the HIF1-α/VEGF pathway Targeting the PI3K/mTOR or HIF-1α pathways could improve radiosensitivity
Chen et al. (2) [48]	2019	CRC	In vitro (cell lines): HCT 116, HT 29, SW480 In vivo (xenograft): HCT-116 + mice	NVP-BEZ235 (maintenance therapy)	PI3K and mTOR inhibitor	Variable range (1–5 Gy)	Enhanced apoptosis Prolonged inhibition of cellular viability Disruption of DSB repair pathways
Djuzenova et al. [49]	2016	CRC	In vitro (cell lines): SW480, SW48	PI-103 (+ NVP-AUY922)	PI3K, mTOR, DNA-PK inhibitor HSP-90 inhibitor	Variable range (0–8 Gy)	Enhanced radiosensitising effect following treatment when treatment started 3 h before radiotherapy and continued for 24 h after radiotherapy
Chen et al. (1) [18]	2015	CRC	In vitro (cell lines): HCT 116, SW 620 (*KRAS* mutant), HT 29 (*KRAS* wild type) In vivo (xenograft): HCT-116 + mice	NVP-BEZ235	PI3K and mTOR inhibitor	Variable range (0–6 Gy)	Dose dependent increase in radiotherapy sensitivity and increased apoptosis after treatment with drug and radiotherapy Loss of expression of proteins responsible for DNA damage repair, cell growth and proliferation Significant reduction in xenograft tumour size
Prevo et al. [23]	2008	CRC	In vitro (cell lines): HCT116, DLD-1	PI-103	PI3K, mTOR and DNA-PK inhibitor	Variable range (0–6 Gy)	Reduced AKT phosphorylation Radiosensitisation independent of PI3K overexpression, *EGFR* and *RAS* mutation status
Manegold et al. [50]	2008	CRC and Pancreatic Cancer	In vitro (cell line): CT-26 (murine CRC) L3.6pl (human pancreatic) In vivo (xenograft): CT-26/L3.6pl + mice	Everolimus	mTOR inhibitor	10 Gy or 20 Gy	Radiosensitising effects observed. Significant reduction in tumour size of pancreatic and CRC cell xenografted mice treated with drug and radiotherapy compared to control (*p* < 0.05)
Park et al. [66]	2017	Pancreatic Cancer	In vitro (cell lines): Miapaca-2, PANC-1 In vivo (xenograft): Miapaca-2 + mice	HS-173	PI3K inhibitor	Variable range (0–10Gy)	G2/M cell cycle arrestM Inhibited Ataxia-Telangiectasia Mutated (ATM) protein and DNA-PKcs Improved radiotherapy response by inhibiting the DNA damage-repair pathways
Dumont et al. [60]	2019	Prostate Cancer	In vivo (xenograft): PC-3 + mice	Rapamycin	mTOR inhibitor	Radioisotope treatment (Lu-labeled GRPr antagonist) Dose—37 MBq (for 72 h)	Rapamycin alone had no effect. With rapamycin and GRPr antagonist, more effective radiosensitisation observed in xenografts
Chang et al. [56]	2014	Prostate Cancer	In vitro (cell line): CAP-RR	BKM120 Rapamycin NVP-BEZ235 PI-103	PI3K Inhibitor mTOR inhibitorPI3K and mTOR inhibitor PI3K, mTOR and DNA-PK inhibitor	6 Gy	Dual inhibitors superior at radiosensitising – increased apoptosis and autophagy Dual inhibition led to G2/M arrest and suppressed DSB repair mechanisms
Potiron et al. [57]	2013	Prostate Cancer	In vitro (cell lines): PC-3, DU 145	NVP-BEZ235	PI3K and mTOR inhibitor	Variable range (0–12.5 Gy)	Radiosensitised both cell lines independent of oxygen concentration or *PTEN* mutation status G2/M arrest
Zhu et al. [58]	2013	Prostate Cancer	In vitro (cell line): PC-3	NVP-BEZ235	PI3K and mTOR inhibitor	Variable range (0–10 Gy)	Radiosensitising effects on PC3 cell line after treatment G2/M arrest
Diaz et al. [59]	2009	Prostate Cancer	In vitro (cell lines): NCI-60 prostate cancer cell line panel In vivo (xenograft): PC-3 + mice	Palomid 529	AKT inhibitor	Variable range (2–8 Gy)	Decreased expression of proteins involved in cell survival and proliferation Xenografts demonstrated greater tumour shrinkage (77.4%) after treatment

**Table 2 cancers-12-01278-t002:** Preclinical studies exploring the role of PI3K, AKT and/or mTOR inhibitors radiosensitising SCC and NSCLC.

Author	Year	Type of Cancer	Experimental Model	Drug (s) Tested	Drug Category	Radiotherapy Dose	Summary Outcome
Assad et al. [79]	2018	Cervical SCC	In vitro: (cell line): HeLa	Temsirolimus, everolimus, resveratrol, curcumin, epigallocatechin gallate	mTOR inhibitors	2 Gy	Radiosensitisation observed with mTOR inhibitors through late apoptosis and necrosis
Yu et al. [37]	2017	Head and Neck (Oral) SCC	In vitro (cell lines): OML1, OML1-R, SCC4, SCC25, Patient derived cell lines In vivo: (xenograft) OML1-R + mice	NVP-BEZ235 Everolimus AZD2014 BKM120	PI3K and mTOR inhibitor mTOR inhibitorm TOR inhibitor PI3K inhibitor	10 Gy or 0–4 Gy (variable)	Dual inhibition of PI3K and mTOR performed significantly better by inhibiting cell proliferation BEZ235 + radiotherapy reduced cell viability 2.4–6.3-fold Radiotherapy with BEZ235 significantly suppressed the growth of xenograft tumours (*p* =0.00017)
Leiker et al. [75]	2015	Head and Neck SCC	In vitro (cell lines): UMSCC1-wtP53, UMSCC46-mtP53, normal human fibroblast line (1522) In vivo (xenograft): UMSCC1 + mice	PF-05212384	PI3K and mTOR inhibitor	Variable range (0 to 8 Gy)	Enhanced radiosensitisation demonstrated in vitro and in vivo after treatment Compared to normal human fibroblasts tumour cells more effectively radiosensitised.
Liu et al. [73]	2015	Head and Neck (Naso-pharyngeal) SCC	In vitro (cell lines): CNE-2, 5-8F, 6-10B, CNE-1, NP69 In vivo (xenograft): 5-8F + mice	GSK2126458 PKI-587	PI3K and mTOR inhibitor PI3K and mTOR inhibitor	4 Gy	Both drugs: Increased DNA damage and G2–M cell cycle delay Induced apoptosis and inhibited cell proliferation Significantly inhibited xenograft tumour growth and proliferation Suppressed phosphorylation of mTOR, AKT and 4E-BP1
Cerniglia et al. [74]	2012	Head and Neck SCC	In vitro (cell line): SQ20B In vivo: (xenograft) SQ20B + mice	NVP-BEZ235	PI3K and mTOR inhibitor	Variable range (0 to 6 Gy)	Knockdown of pathway components AKT, p110- α, or mTOR led to radiosensitisation, but not to the same extent as NVPBEZ235 Loss of resolution of H2A histone family member X foci (_γ_-H2AX) Induced autophagy in cell lines and xenograft tumours
Fokas et al. (1) [77]	2012	Head and Neck (laryngeal and hypo -pharyngeal) SCC	In vitro (cell lines): SQ20B, FaDu	NVP-BGT226 NVP-BEZ235	PI3K and mTOR inhibitor PI3K and mTOR inhibitor	6 Gy	Both inhibitors can enhance radiation-induced killing of tumour cellsBoth inhibited phosphorylation of AKT, mTOR and led to DNA damage persistence (increased _γ_-H2AX foci) G2 cell cycle delay
Bozec et al. [78]	2011	Head and Neck SCC	In vivo (xenograft): CAL33 + mice	Temsirolimus (+ cetuximab and bevacizumab)	mTOR inhibitor Cytotoxic chemotherapy + VEGF inhibitor	6 Gy three times a week	Longest delay in tumour growth observed when temsirolimus combined with cetuximab, bevacizumab and radiotherapy (*p*-0.01) Reduced Ki-67 and BCL2 implying decreased tumour proliferation as well as anti-apoptotic effects
Fokas et al. (2) [76]	2011	Head and Neck SCC	In vitro (cell line): FaDU HRE-Luc In vivo (xenograft): FaDU HRE-Luc + mice	NVP-BEZ235BKM120	PI3K and mTOR inhibitor PI3K inhibitor	6 Gy	Dual inhibitor modulated the tumour microenvironment Radiosensitised tumours by prolonging the time taken for normalisation of tumour vasculature
Prevo et al. [23]	2008	Head and Neck SCC	In vitro (cell line): SQ20B	PI-103	PI3K, mTOR and DNA-PK inhibitor	Variable range (0–6 Gy)	Reduced AKT phosphorylation. Persistent DNA damage after treatment (increased _γ_-H2AX foci). G2/M phase delay. Overall effects led to reduced cell survival.
Holler et al. [39]	2016	NSCLC	In vitro (cell lines): H661, H460, SK-MES-1, HTB-182, A549	Rapamycin MK-2206	mTOR inhibitor AKT inhibitor	Variable range (0–5 Gy)	AKT inhibition led to rapamycin induced radiosensitisation in radio-resistant NSCLC cells Dual inhibition of AKT and mTOR significantly inhibited DNA DSB repair compared to single inhibition
Toulany et al. [38]	2016	NSCLC		PI-103 PD98059	PI3K, mTOR and DNA-PK inhibitor MEK inhibitor	4 Gy	Decreased DNA DSB repair by inhibited DNAP-PKcs activity short term Combining MEK inhibitor prevented indirect activation of AKT (RAS/RAF/MEK pathway). This led to more prolonged and greater radiosensitisation effects
Kim et al. (1) [80]	2014	NSCLC	In vitro (cell line): H460-Luc2 (cisplatin resistant clone) In vivo (xenograft): H460-Luc2 (cisplatin resistant clone) + mice	NVP-BEZ 235	PI3K and mTOR inhibitor	Variable range (0–6Gy)	BEZ-235 enhanced radiosensitivity (*p*-0.01) Increased autophagy and cell proliferation Significant tumour growth delay observed in treated xenografts Reduced caspase-3 activity, cell proliferation and vascular density
Kim et al. (3) [81]	2013	NSCLC	In vitro (cell lines): H1650, HCC827	Everolimus	mTOR inhibitor	Variable range (0–3Gy)	Enhanced autophagy following mTOR inhibition and radiotherapy Radioresistance observed in HCC827 cell lines
Mauceri et al. [82]	2012	NSCLC	In vitro (cell line): A549 In vivo (xenograft): A549 + mice	Everolimus	mTOR inhibitor	5 × 6 Gy	Everolimus altered gene expression in treated cells Everolimus with radiotherapy significantly slowed tumour growth compared to radiotherapy alone in xenograft (*p* = 0015).
Konstantinidou et al. [36]	2009	NSCLC	In vitro (cell lines): H23, H460, H2122 In vitro (mouse cell lines): LKR10, LKR13 In vivo (xenograft): H460 + mice	NVP-BEZ235 LY294002 + rapamycin	PI3K and mTOR inhibitor PI3K and PI3K-like kinase inhibitor + mTOR inhibitor	Variable range (1–6 Gy)	Anti-proliferative effects, G1 growth arrest and overall radiosensitisation observed in all treated cell lines and xenografts NVP-BEZ 235 or LY294002 + rapamycin demonstrated comparable radiosensitising effects
Kim et al. (2) [83]	2008	NSCLC	In vitro (cell line): H460 In vivo (xenograft): H460 + mice	Everolimus Z-DEVD	mTOR inhibitor Caspase -3 inhibitor	Variable (0–6Gy)	The combination of Z-DEVD and RAD001 more potently radiosensitised H460 cells than individual treatment alone. Increased radiosensitisation predominantly through enhanced autophagy

**Table 3 cancers-12-01278-t003:** Preclinical studies exploring the role of PI3K, AKT and/or mTOR inhibitors radiosensitising GBM.

Author	Year	Type of Cancer	Experimental Model	Drug (s) Tested	Drug Category	Radiotherapy Dose	Summary Outcome
Shi et al. [98]	2018	GBM	In vitro (cell lines): A172, SHG44, and T98G In vivo (xenograft)	NVP-BEZ235	PI3K and mTOR inhibitor	2 Gy	Higher p27 and lower Bcl-2 expression in cells treated with radiotherapy, temozolomide and NVP-BEZ235 Inhibited tumour growth and prolonged survival with combined treatment all three modalities
Djuzenova et al. [49]	2016	GBM	In vitro (cell lines): GaMG, SNB19	PI-103	PI3K, mTOR and DNA-PK inhibitor	0, 2 Gy or 8 Gy	Enhanced radiosensitising effect following treatment Enhanced radiosensitisation to heat shock protein 90 inhibitor NVP-AUY922
Choi et al. [94]	2014	GBM	In vitro (cell lines): U251, U87, T98G	PI-103 Rapamycin	PI3K, mTOR and DNA-PK inhibitor mTOR inhibitor	Variable range (0–8 Gy)	Radiotherapy with drug increased cytotoxic effects No discernible radiosensitising effect with rapamycin
Del Alcazar et al. [90]	2014	GBM	In vitro (cell line): U87MG In vivo (xenograft): GBM9 + mice	NVP-BEZ235	PI3K and mTOR inhibitor	0, 2 Gy or 10 Gy	Reduced expression of DNA-PKcs and ATM kinase Attenuated repair of radiotherapy induced DNA damage Sensitized tumours to temozolomide and radiotherapy
Kuger et al. [2,91]	2013	GBM	In vitro (cell lines): GaMG (*PTEN* wt, *p53* mut), DK-MG (*PTEN* wt, *p53* wt), U373 (*PTEN* mut, *p53* mut), U87-MG (*PTEN* mut, *p53* wt)	NVP-BEZ235 Schedule 1: 24hr before radiotherapy Schedule 2: 1 h before and for 48hr post radiotherapy	PI3K and mTOR inhibitor	Variable range (0–8 Gy)	Enhanced radiosensitivity under schedule 2 as opposed to schedule one Protracted DNA repair Prolonged G2/M arrest and apoptosis
Wang et al. [92]	2013	GBM	In vitro (cell line): SU-2 cells (Glioma stem cells)	NVP-BEZ235	PI3K and mTOR inhibitor	Variable range (0–8 Gy)	Autophagy, increased apoptosis Decreased DNA repair capacity G1 cell cycle arrest
Cerniglia et al. [74]	2012	GBM	In vitro (cell line): U251MG	NVP-BEZ235	PI3K and mTOR inhibitor	Variable range (0–6 Gy)	Radiosensitises cells and induces autophagy
Mukherjee et al. [93]	2012	GBM	In vitro (cell lines): U251, U118, LN18, T98G, LN229, SF188, 1BR3, AT5, M059K, M059 In vivo (xenograft): U87 + mice	NVP-BEZ235	PI3K and mTOR inhibitor	0, 2 Gy or 10 Gy	Reduced expression of ATM and DNA-PKcs G2/M Arrest
Li et al. [95]	2009	GBM	In vitro (cell line): U87MG	SH-5 MK-2206 LY294002	AKT inhibitor AKT inhibitor PI3K and PI3K-like kinase inhibitor	Variable range (0–9 Gy)	Enhanced radiosensitivity noted with all three agents
Kao et al. [96]	2007	GBM	In vitro (cell line): U251MG	LY294002	PI3K and PI3K-like kinase inhibitor	Variable range (0–6 Gy)	Enhanced radiosensitivity following treatment
Nakamura et al. [97]	2005	GBM	In vitro (cell line): U251 MG	LY294002 Rapamycin	PI3K and PI3K-like kinase inhibitor mTOR inhibitor	Variable range (0–8 Gy)	Low doses of LY294002 sensitized U251 MG to clinically relevant doses of radiation No effect observed with rapamycin
Shinohara et al. [99]	2005	GBM	In vitro (cell lines): GL261, endothelial cell line In vivo (xenograft): GL261 + mice	Everolimus	mTOR inhibitor	Variable range (0–7 Gy)	Reduced tumour growth by enhanced radiosensitisation of the tumour vascular endothelium
Eshleman et al. [100]	2002	GBM	In vitro (cell lines): U87, SKMG-3 In vitro (spheroids): U87 In vivo (xenograft): U87 + mice	Rapamycin	mTOR inhibitor	Variable range (0–10 Gy)	Rapamycin had no effect on cell lines However, the drug significantly enhanced the response to radiotherapy on xenograft and spheroids following treatment

**Table 4 cancers-12-01278-t004:** Preclinical studies exploring the role of PI3K, AKT and/or mTOR inhibitors radiosensitising other types of cancer.

Author	Year	Type of Cancer	Experimental Model	Drug (s) Tested	Drug Category	Radiotherapy Dose	Summary Outcome
Detti et al. [106]	2016	Renal cell carcinoma (RCC)	Human patient case-report	Everolimus	mTOR inhibitor	Total dose = 20 Gy	RCC vertebral metastases treated successfully with radiotherapy and everolimus
Liu et al. [107]	2014	Hepatocellular Carcinoma	In vitro (cell lines): Huh7, BNL	BKM120 Rapamycin	PI3K inhibitor mTOR inhibitor	Variable range (0–10 Gy)	Both drugs promoted apoptosis and reduced DSB break repair to a certain extent Dual inhibition with BKM120 and rapamycin significantly enhanced radiosensitivity of treated cells
Qiao et al. [108]	2013	Human Burkitt’s Lymphoma	In vitro (cell lines): Namalwa, Ramos, Raji	LY294002 SH-5	PI3K and PI3K-like kinase inhibitor AKT inhibitor	5Gy	Enhanced apoptosis following treatment with the pathway inhibitors followed by radiotherapy
Fokas et al. (1) [77]	2012	Bladder (TCC)	In vitro (cell line): T24	NVP-BEZ235	PI3K and mTOR inhibitor	Variable range	Dual inhibitors enhance radiation-induced killing of endothelial cells
		Endothelial cells	In vitro (cell line): HUVEC, HDMVC	NVP-BGT226	PI3K and mTOR inhibitor	(0–6 Gy)	
Fokas et al. (2) [76]	2011	Sarcoma	In vitro (cell line): HT-1080 HRE-Luc In vivo (xenograft): HT-1080 + mice	NVP-BEZ235 BKM120	PI3K and mTOR inhibitor PI3K inhibitor	6 Gy	Dual inhibitor modulated the tumour microenvironment Radiosensitised tumours by prolonging the time taken for normalisation of tumour vasculature
Murphy et al. [105]	2009	Sarcoma Endothelial cells	In vitro (cell lines): SK-LMS-1 leiomyosarcoma, HT-1080 fibrosarcoma, SW-872 liposarcoma cells, In vitro (cell line): HDMEC	Rapamycin	mTOR inhibitor	Variable range (0–6 Gy)	Rapamycin radiosensitised all cell lines in vitro
Prevo et al. [23]	2008	Sarcoma Bladder (TCC)	In vitro (cell line): HT-1080 T24	PI-103	PI3K, mTOR and DNA-PK Inhibitor	Variable (0–6 Gy)	Persistent DNA damage after treatment G2/M phase arrest
Manegold et al. [50]	2008	Endothelial cells	In vitro (cell line): HUVEC	Everolimus	mTOR inhibitor	10 Gy or 20 Gy	Endothelial cells most sensitised to radiotherapy by mTOR inhibition

**Table 5 cancers-12-01278-t005:** Clinical trials exploring the role of PI3K, AKT and/or mTOR inhibitors radiosensitising various cancer.

Author	Year	Type of Cancer	No. of Patients	Stage of Disease	Experimental Model	Drug (s) Tested	Drug Category	Radiotherapy Dose	Summary Outcome
de Melo et al. [112]	2016	Cervical Cancer	13	Primary stage IIB, IIIA or IIIB	Phase-I clinical trial	Everolimus + cisplatin + radiotherapy	mTOR inhibitor with traditional chemoradiotherapy	External Beam 4500 cGy, 25 fractionsBrachytherapy: 2400 cGy, 4 insertions	Maximum tolerated dose (MTD) of everolimus with radiotherapy and cisplatin is 5 mg/day
Chinnaiyan et al. [103]	2018	GBM	171	Newly diagnosed	Phase-II clinical trial	Everolimus (10mg/day) + temozolomide (TMZ) + radiotherapy	mTOR inhibitor with traditional chemoradiotherapy	60 Gy in 30 fractions of 2 Gy each	Increased toxicity. Overall survival of everolimus group worse (*p* = 0.008)
Ma et al. [102]	2015	GBM	100	Newly diagnosed	Phase-II clinical trial	Everolimus (70mg/wk) + temozolomide + radiotherapy	mTOR inhibitor	Total dose: 60 Gy	Moderate toxicity observed following everolimus radiotherapy and TMZ No appreciable survival benefit over control
Wen et al. [104]	2015	GBM	54	Newly diagnosed	Phase-I clinical trial	XL 765 (30–90 mg once daily or 20–50 mg twice daily) + temozolomide + radiotherapy	PI3K and mTOR inhibitor	Total dose of 60 Gy (1.8–2 Gy a day, 5 days a week)	Drug + temozolomide with or without radiotherapy feasible with favourable safety profile Moderate level of PI3K/mTOR pathway inhibition observed on skin biopsies MTD = 90 mg/day
Sarkaria et al. [101]	2011	GBM	18	Newly diagnosed	Phase-I clinical trial	Everolimus (30 mg/wk or 50 mg/wk or 70 mg/wk) + temozolomide + radiotherapy	mTOR inhibitor	Total dose: 60 Gy in 30 fractions	Favourable safety profile following TMZ + radiotherapy Changes in tumour metabolism following everolimus MTD = 70 mg/wk (10 mg/day)
NCT00858663	-	Head and Neck SCC	Not known	Not known	Phase-I	Everolimus	mTOR inhibitor	Not known	N/A
NCT02113878	-	Head and Neck SCC	Not known	Not known	Phase-Ib	BKM120	PI3K inhibitor	Not known	N/A
Deutsch et al. [87]	2015	NSCLC	26	Primary NSCLC (Stage III-IV)	Phase-I clinical trial	Everolimus	mTOR inhibitor	Median total dose of 66 Gy (range, 28–66), Median number of 33 fractions (range, 14–33)	Treatment feasible and safe Authors recommend close observation for pulmonary toxicity
NCT00374140 (In progress)	-	NSCLC	Not known	Not known		Everolimus	mTOR inhibitor	Not known	N/A
NCT02128724 (In progress)	-	NSCLC	Not known	Not known		BKM120	PI3K inhibitor	Not known	N/A
Azria et al. [61]	2017	Prostate Cancer (high risk locally advanced)	14	Primary locally advanced non metastatic prostate cancer (≥T3, Gleason score ≥ 8)	Phase-I clinical trial	Everolimus (5 or 7.5 or 10 mg)+hormone therapy + Radiotherapy	mTOR inhibitor	74 Gy in 37 fractions of 2 Gy	Everolimus was tolerated with hormone and radiotherapy with minimal side effects MTD = 7.5 mg/day. Recommended dose for phase-II studies is 5mg/day
Narayan et al. [62]	2017	Prostate Cancer (recurrent)	18	Biochemical recurrence following prostatectomy	Phase-I clinical trial	Everolimus (5 or 7.5 or 10 mg) + radiotherapy	mTOR inhibitor	66.6 Gy in 37 fractions of 1.8 Gy	Everolimus dose of ≤10mg/day is safe and toleration in combination with radiotherapy
Gelsomino F [52]	2017	Rectal Cancer	12	Primary resectable rectal cancer (T3-4, N0-2)	Phase-I/II clinical trial (n = 12)	Everolimus (2.5 or 5 or 7.5 or 10 mg) + 5FU + radiotherapy	mTOR inhibitor with traditional chemoradiotherapy	1.8 Gy/fraction 50.4 Gy in 28 daily fractions, 5 days/week	No increase in toxicity at any of the doses with 5-FU and radiotherapy No significant increase in pCR MTD—10 mg
Buijsen et al. [51]	2015	Rectal Cancer	13	Primary resectable rectal cancer (T2-3, N0-1)	Phase-I/II clinical trial Phase-I (n = 13) Phase-II (n = 31)	Rapamycin (2 mg or 4 mg or 6 mg) + radiotherapy	mTOR inhibitor	5 × 5 Gy	Neoadjuvant radiotherapy and rapamycin feasible with no significant increase in toxicity (MTD = 6mg) No increase in post-operative complications rates after delaying duration to surgery to 6 weeks Significant reduction in tumour metabolic activity but pCR rate was still 10% within small sample sized study

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
