# Peer review of "Radiosensitising Cancer Using Phosphatidylinositol-3-Kinase (PI3K), Protein Kinase B (AKT) or Mammalian Target of Rapamycin (mTOR) Inhibitors"

_cancers, 2020, doi:10.3390/cancers12051278_

Round 1

Reviewer 1 Report

Overall this is a comprehensive and good quality review. 

  1. It would be helpful to include data on the role of PI3k/mTOR inhibitors as radiosensitizers in the treatment of lymphomas. Are there any preclinical data to suggest there may be an effect?
  2. Line 374 mentions pancreatic cancer, but the data is not included in Table 4. 
  3. Table 5 - it would be helpful to include a column with the number of patients treated in each of those trials
  4. Remove semicolon in lines 66, 73
  5. Line 86 and 87 - sentence "The net effect..." needs grammatical correction
  6. Line 124 - italicize in vitro and in vivo 
  7. Line 232 - may want to reword as "A phase-I clinical trial of 15 patients with locally advance PC..."
  8. Line 252 - remove "and radiotherapy"
  9. Tables 1-4 - instead of "Research exploring..." may want to rephrase as "Preclinical studies exploring..."

Author Response

Dear Sir/Madam

Thank you very much for taking the time to review our article. Please find the attached responses.

Kind Regards

The Authorship

Reviewer 2 Report

This is a comprehensive review about the strategy of radiosensitising cancer by using the inhibitors of PI3K/AKT/mTOR pathway in a variety of cancers. This manuscript is interesting clinically and importantly. However, there are some concerns listed below.

  1. The references about the in vitro studies only provided in this manuscript are advised to be deleted.  In vivo studies or in viro plus in vivo studies are more scientific soundness.
  2. Please list the treatment purpose, treatment naive cancer or recurrent metastasis cancer about the clinical trials in table 5. 

Author Response

Dear Sir / Madam

Thank you for taking the time to review our article. Please find the attached responses. 

Kind Regards

The Authorship

Reviewer 3 Report

Dear Authors,

the review is well organized and clear.

The informations are interesting and logically described.

There are some minors aspects that should be improve.

1-Figure 1: the resolution is very low. Probably the image needs an all-out layout.

In my opinion could be a good idea insert the PI3K/Akt/mTOR inhibitors in a cartoon against their target.

2- Tables 1,2,3,4, and 5 should be differently organized. In order to make the tables more intuitive, the inhibitors should be grouped by type of target. 

In conclusion, the review deserve to be accepted by that journal.

Author Response

Dear Sir/Madam

Thank you for taking the time to review our article. Please find the attached responses.

Kind Regards

The Authorship

This manuscript is a resubmission of an earlier submission. The following is a list of the peer review reports and author responses from that submission.